# Environmental Regulation Promotes Eco-Efficiency through Industrial Transfer: Evidence from the Yangtze River Economic Belt in China

**DOI:** 10.3390/ijerph191610127

**Published:** 2022-08-16

**Authors:** Yelin Dai, Yue Liu, Xuhui Ding, Chundu Wu, Yu Chen

**Affiliations:** 1School of Finance and Economics, Jiangsu University, Zhenjiang 212013, China; 2School of Geography, Nanjing Normal University, Nanjing 210042, China; 3School of Information Science and Technology, Nanjing Normal University Zhongbei College, Zhenjiang 212300, China

**Keywords:** environmental regulation, industrial transfer, ecological efficiency, Markov process

## Abstract

How does environmental regulation affect ecological efficiency? What is the role of industrial transfer in the mechanism of action? Relations and interactions between the three determine economic quality when ecological performance is concerned. Empirical studies in this paper are based on samples from the Yangtze River economic belt in China, which contributes nearly half of China’s GDP. By measuring environmental regulation, industrial transfer, and ecological efficiency, data and indexes are prepared for investigating the driving mechanism of environmental regulation and illustrating of the role of industrial transfer. By applying the Markov process to model industrial transfer between regions, the dynamic of transfer is simulated and facilitates further study on the effects of industrial transfer. Finally, this paper concludes that by targeting on the improvement of ecological efficiency, environmental regulation releases its utility through industrial transfer. The highlights include three aspects. Theoretically, it illustrates the driving mechanism of improving the eco-efficiency by environmental regulation. Technically, it pioneers a methodology for describing the regional industrial transfer by modeling it with a Markov process. Practically, the conclusion supplies insights into the inherent law of sustainable development for policy makers.

## 1. Introduction

To illustrate the mechanism of environmental regulation, industrial transfer, and ecological efficiency, we propose to select a typical sample region that should be wide and economically significant to investigate the triple aspects. As well known, the Yangtze River economic belt is the most important inland river economic belt in China. It gathers 42.8% of China’s population and 44.1% of GDP with only 21.4% of China’s land area. As the focus and vitality of China’s economy, the Yangtze River economic belt will become a demonstration of regional high-quality development. Over the past decade, state and local governments have guided the orderly transfer of resource and labor-intensive and domestic demand-oriented capital and technology-intensive industries to the middle and upper reaches. It cannot be ignored that the related industries in the eastern coastal areas have also been transferred to the central and western regions due to problems such as labor and raw materials, which is also in line with the theory of industrial gradient transfer. As noted in the pollution shelter theory, the pollution industry is transferred to underdeveloped areas with the improvement of environmental regulation in developed areas, and the pollution industry is transferred upstream to the middle and upper reaches of the Yangtze River economic belt. The environmental Kuznets curve also puts forward that there is an inverted U-shaped relationship between eco-environmental quality and economic development level, and the industrial gradient transfer in the Yangtze River Basin is just in line with this objective law, but the pollution brought by the clip is more hidden. Therefore, as for the Yangtze River economic belt, the relationship and interaction among environmental regulation, industrial transfer, and ecological efficiency have become important economic research topics.

Based on the environmental and industrial samples of the Yangtze River economic belt, and with respect to the interaction mechanism of environmental regulation, industrial transfer, and ecological efficiency, this paper focuses on the impact of the first two on ecological efficiency. Intuitively, both environmental regulation and industrial transfer will promote ecological benefits, and this paper is mainly committed to revealing the impact of environmental regulation on ecological efficiency by the intermediate hand, which indeed is the industrial transfer. Firstly, this paper measures the environmental regulation, industrial transfer, and ecological efficiency of provinces and cities in China’s Yangtze River economic belt. Because the data of industrial transfer are only limited to the amount of industrial transfer in and out of provinces and cities, there is a lack of specific values of industrial transfer between provinces and cities. Therefore, this paper introduces the Markov process to simulate the dynamics of industrial transfer between provinces and cities, and on this basis, numerically estimate the effect of industrial transfer on ecological efficiency, so as to reveal the role of environmental regulation in promoting ecological efficiency through industrial transfer and provide theoretical support for policy suggestions.

The remaining parts are arranged as follows. In Section 2, we review the existing research on related theory and topics. In Section 3, measurements of environmental regulation, industrial transfer, and ecological efficiency are established after data collection and selection. Section 4 constructs the model of industrial transfer driven by environmental regulation, based on which we proceed with the numerical implementation of industrial transfer driven by environmental regulation, which is provided in Section 5. Hence, in Section 6, the eco-efficiency driving mechanism of environmental regulation is verified. All of the conclusions established and the management implications achieved in the preceding sections are summarized in Section 7.

## 2. Literature Review

Relevant research on the three aspects, namely, environmental regulation, ecological efficiency, and industrial transfer, is summarized in this section, especially for the developments and advances of each topic. Additionally, perspectives about their correlations are presented and integrated based on the existing research articles.

### 2.1. Research on Environmental Regulation

In terms of environmental regulation tools, many experts generally divide them into command-and-control policy tools, market-oriented policy tools, voluntary environmental policy tools, etc. Some experts also divide them into formal and informal environmental policy tools according to the implementation mechanism of environmental policy tools. The theory and practice of environmental policy tools originated from Western developed countries. In recent years, many innovations have been made in environmental policy tools for specific problems, and a variety of environmental policy tools have been continuously integrated and used. Ref. [1] mainly explored the role played by stakeholders through negotiation tools and the effectiveness of various types of environmental protection agreements in environmental governance. Ref. [2] proposed quality control of environmental policy evaluation tools based on specific cases in the Netherlands and Denmark, and advocated that the government should be responsible for enhancing the effectiveness of evaluating environmental policy tools. Ref. [3] selected the sustainable tool of environmental policy, namely, the social ecological model, adopted sensitivity analysis and other system dynamics methods to consider environmental policy assessment as a whole, and emphasized environmental impact assessment as a powerful tool. Ref. [4] applied the integration of environmental policies as the entry point to explain multilevel governance issues at the EU level, explored how to transform abstract policy concepts into practical policy tools, and focused on knowledge creation and authorization in the multilevel system.

Many scholars also select different environmental regulation variables for different research objects to conduct relevant empirical studies, and mostly use relevant data in annual reports for some microenterprises. Ref. [5] collected relevant laws and regulations to evaluate India’s environmental regulations from the perspective of air and water pollution and their corresponding environmental regulations, and verified the effectiveness of air and water control. Ref. [6] adopted the indicator referred as PACE, which is the expenditure for pollution reduction and control. It also defines the intensity of environmental regulation from the perspective of expenditure, which is different from some measures based on the achievements of environmental regulation. Ref. [7] adopted environmental taxes to represent the severity of environmental regulations on carbon emissions in the European Union.

### 2.2. Research on Ecological Efficiency

Corporate carbon performance is a newly developed concept to measure the achievement effect obtained by carbon consumption. Traditionally, corporate carbon performance is defined as a ratio between a certain financial indicator (such as the operation revenue) and the amount of carbon emission [8,9]. Based on the principle of simplicity, one of the widely applied expressions of carbon performance for Chinese enterprises with carbon emission is a ratio of sales and carbon emission, as used in [10]. Other approaches, such as input and output models with nonradial directional distance, are available to calculate the carbon performance [11,12,13]. From the perspective of corporate management, carbon performance is well applied to investigate the impact of carbon emission and green innovation on the financial performance.

Many scholars continue to develop and innovate the concept of ecological efficiency on an early basis, and also explore and improve the evaluation system of ecological efficiency, trying to be closer to the reality of ecological construction. Ref. [14] compared and analyzed the ecological efficiency presented by DEA in African countries based on the affirmation of ecological footprint and biocapacity. Ref. [15] adopted the entropy weight method to measure provincial ecological efficiency in China, and used the coupling coordination degree model to evaluate the coordination relationship among ecological efficiency, natural resources, and financial development. Ref. [16] evaluated the economic–ecological efficiency of six regions in Azerbaijan based on solar energy resources, technology, economy, market, and other factors. Ref. [17] proposed a sustainable development index to replace the human development index, which was used to represent the ecological efficiency of human development. Key ecological variables such as carbon dioxide and material footprint should be retained in the calculation of the formula. Ref. [18] collected CLC data sets and used FRAGSTATS and Statistica to evaluate the ecological efficiency of Polish landscape conservation. Ref. [19] used the cross-efficiency model of the Shannon entropy index and data envelopment analysis DEA to rank the ecological efficiency of Italian cities.

Scholars extend the application scope of eco-efficiency to various fields, and also adopt different research methods and analysis models due to differences in application fields. Ref. [20] proposed ecological efficiency of foreign trade to determine the advantages of foreign trade. Ref. [21] used input-oriented data envelopment analysis to measure and decompose the economic, environmental, and ecological efficiency levels of agricultural production in OECD countries. Starting from the ecological modernization of the electric power industry, ref. [22] selected 437 fossil fuel power plants in the United States to conduct an ecology–technical efficiency investigation, and evaluated whether their organizational and technological innovation could improve such joint efficiency. Ref. [23] mainly considered the limitations and difficulties of the concept of ecological efficiency and reconsidered the economic logic of ecological modernization. Ref. [24] proposed a new indicator to evaluate the sustainable development of a country. The concept of ecological efficiency was put forward from the perspective of ecological footprint and ecological reserve, and the specific evaluation method of ecological efficiency was put forward.

### 2.3. Research on Industrial Transfer

Since the 1990s, research on industrial transfer has gradually turned to empirical research and began to consider specific countries, specific industries, and specific enterprises. Some experts mainly studied the spatial distribution of industrial transfer to explore the transfer strategies of industry and knowledge under the challenge of globalization and digitalization. Ref. [25] even explained industrial transfer from the perspective of vertical specialization in north–south trade, and proposed that the decline of trade costs in different production stages would encourage vertical specialization, and final product production would shift to southern countries under a certain threshold value.

In terms of the mechanism and driving factors of industrial transfer, domestic and foreign experts also give explanations from different aspects, and combine specific industries and micro enterprises to explain and test. Ref. [26] explained the problem of industrial transfer from the perspective of regional clusters on a global scale, and proposed that more regional clusters are the potential driving force for the reorganization of global production activities caused by global outsourcing or overseas outsourcing. Ref. [27] proposed that the benefits brought by an agglomeration economy promote the concentration of industries in cities from the perspective of business location selection and relocation decisions in the United States.

### 2.4. Correlation Study of the Three Aspects and Summation of the Literature Review

In terms of the relationship between environmental regulation and industrial transfer, although environmental regulation is intended to promote energy saving and emission reduction and industrial upgrading, it may objectively cause inter-regional transfer of high-energy consumption and high-pollution industries because of the level discrepancy of interregional environmental regulations. Those industries or individual enterprises may be driven toward areas with less strict regulations. Ref. [28] conducted a field study on environmental regulation and health protection in asbestos production in Japan, Germany, Indonesia, and South Korea, aiming at the process of asbestos industry transfer in Asia and the environmental health problems caused by it. Ref. [29] believe that polluting industries will move to countries with less stringent environmental regulations, but pollution intensity and ease of relocation will affect the response of various industries to strict environmental regulations. Ref. [30] conducted a case study of industrial enterprises in Algeria to discuss the impact of market forces and environmental regulations on industrial pollution, especially the transfer of cleaner production technology between developed industrial countries and developing countries.

Concerning the impact of industrial transfer on the ecological environment, the main theoretical hypothesis is the pollution paradise hypothesis. The pollution paradise hypothesis, also referred to as the pollution refuge hypothesis, was first proposed by [31]. This hypothesis has also been analyzed and verified in the process of industrial transfer between countries, and has also been applied to inter-regional transfer within countries. Ref. [32] used multivariate framework analysis to test the effectiveness of the Gulf States’ pollution haven hypothesis and found that energy consumption and GDP growth were still the main pollution sources in the Gulf States. Ref. [33] investigated the relationship between FDI inflow and CO_2_ emissions in Turkey by using data from more than 40 years, and proposed that such one-way effect supports the pollution refuge hypothesis in Turkey.

In terms of the relationship between environmental regulation and ecological environment, the original intention of environmental regulation is to protect the ecological environment and other good intentions, but good intentions do not always bring good results, the Porter hypothesis or green paradox and other related theories have been constantly developed and tested by many empirical tests. Ref. [34] believed that environmental regulation itself was unnecessary, and pollution emissions would automatically decrease with the continuous consumption of natural resources. Ref. [35] first proposed the concept of green paradox, which intended to restrict the implementation of climate change policies but led to accelerated exploitation of fossil energy and accelerated greenhouse gas emissions, which in turn lead to environmental deterioration. Ref. [36] further divided the green paradox into the weak version and the strong version. The weak version emphasizes those imperfect climate policies that increase short-term carbon emissions, while the strong version emphasizes enhancing the net present value of future losses of climate change. Ref. [37] proposed the Porter hypothesis, which is different from the green paradox, and believed that environmental protection policies in the true sense would not increase the cost of enterprises, but would also enhance competitiveness through the net income generated by innovation. However, some experts believe that Porter’s hypothesis is not valid. Empirical studies of some countries or industries are selected, and they believe that the relationship between environmental regulation and technological innovation is uncertain, and the promoting effect and inhibiting effect coexist, so Porter’s hypothesis is difficult support by empirical evidence. Ref. [38] use empirical evidence from Asian economies to verify the negative impact of carbon tax rate on the investment decisions of enterprises.

To summarize, scholars have conducted detailed discussions on the origin and development of environmental regulations. They can also conduct comparative analysis for different types of environmental regulations and conduct empirical analysis on environmental regulations by using the single or comprehensive indicator method. Existing studies have been able to sort the causes and manifestations of industrial transfer in detail. Combined with several typical industrial transfers in the world, domestic scholars have also explored inter-regional industrial transfers in economic belts or urban agglomerations. In terms of ecological efficiency measurement, many experts use different measurement methods from their own fields to conduct empirical analysis in combination with specific regions, and obtain some spatiotemporal differentiation rules of ecological environmental efficiency. As for the internal relationship between environmental regulation, industrial transfer, and ecological efficiency, scholars have realized that the transfer of polluting industries is caused by environmental regulation, the nonlinear relationship between environmental regulation and economic development stage, and the necessity and difficulty of collaborative environmental governance. Lacking research on an accurate definition of ecological efficiency, however, means failure to accurately understand the law of industrial transfer and pollution industry transfer. It also leads to failure in comprehending environmental regulation or comparing and choosing different types of multiple evaluation systems, and fails to clarify how environmental regulation causes pollution industry transfer, thus affecting the economic belt or the urban ecological environment. It also fails to fully recognize the ultimate impact of local government’s measures to protect the ecological environment on ecological efficiency, which may be inconsistent with traditional theories. This paper attempts to reveal the impact of environmental regulation on ecological efficiency through industrial transfer by providing a vivid depiction of industrial transfer, so as to fill the gap in previous studies.

## 3. Index Selection and Data Collection

To investigate the interaction mechanism of environmental regulation, industrial transfer, and ecological efficiency, we need to quantitatively measure the three aspects. In the sequel, we proceed to establish some index variables indicating the performance of these aspects, and then supply them with real data from China’s Yangtze River economic belt, acquired for the period 2005 to 2018.

### 3.1. Measurement of Environmental Regulation

Generally, observation variables such as pollution control input and pollutant discharge are used to represent the intensity of environmental regulation. They are also divided into cost-based and investment-based environmental regulation according to the nature of capital. According to the statistical caliber of China’s relevant statistical yearbook, only the total investment in environmental pollution control is close to the real performance of environmental regulations. In general, scholars mainly observe and examine pollution control investment from the aspects of proportion, operating cost of pollution control facilities, per capita income level (endogenous environmental regulation intensity), times of inspection and supervision by environmental agencies, and pollution emissions. The proportion of total investment in industrial pollution control in industrial added value is commonly used as a single indicator, because this indicator represents the efforts and governance determination of local governments in environmental regulation from the perspective of economic expenses, and then focuses on the development trend and spatial differentiation of environmental regulation in the Yangtze River economic belt from 2005 to 2018. Owing to the slow updating of this block of data in the yearbook, some of the data from 2019 and 2020 cannot be obtained for the time being. In the empirical link of this paper, data from 2018 and previous years are uniformly adopted. The intensities of environmental regulation in provinces and cities of the Yangtze River economic belt are listed in Table 1.

### 3.2. Measurement of Industrial Transfer

According to the analysis of classical criteria, there are differences in resource endowment, comparative advantages between the Eastern and Western regions, unbalanced distribution of market size demand, and central and local governments have issued a series of policies to support industrial transfer. Of course, some empirical studies have shown that China has not seen large-scale industrial transfer so far, but inter-regional industrial transfer is different from international industrial transfer, and does not need to involve tariffs, labor mobility, or other restrictions. Many scholars insist that industrial transfer is inevitable and happening, and accurate measurement of the scale of industrial transfer is crucial to solving this problem. Some scholars use relative indexes such as regional output value or the proportion of added value to measure industrial transfer. Some experts use the Gini coefficient, Theil index, Herfindahl index, or other methods to measure industrial transfer, which can effectively overcome the influence of relative measure on the size of administrative divisions and reflect the change in industrial value-added degree and the overall process of industrial transfer. The industrial gradient coefficient is also used to indirectly reflect the degree of industrial transfer, and the multiregional input–output model based on inter-regional input–output is often introduced into the measurement process. However, the data of inter-provincial input–output is published only once every five years, so it is not suitable in our analysis for the measurement of annual industrial transfer.

There are 11 provinces and cities in the Yangtze River economic belt, but is not a closed economic entity. Industrial transfer may also happen in forms of export or import between the Yangtze River economic belt and other parts of China. Therefore, we consider 12 regions, among them the last is the external part but still within China.

The deviation share analysis method concerns the industrial transfer as an event. Before the transfer, industrial development was relatively stable, while after the transfer, industrial development has greatly changed, and the relative change before and after the transfer is the scale of industrial transfer. In general, shift-share analysis is a realistic way to measure industrial transfer according to share changes. It can decompose the changes in economic variables for specific regions and describe the transfer of industries or polluting industries across provinces as a whole.

Deviation share analysis generally decomposes the regional increment into separate components: share (national growth component), structural deviation, and competitiveness deviation, among others (see Formula (1) below), which are the values in the early stage of the regional primary industry, the values in the late stage of the industry, and the change value, respectively. National growth component refers to the study of industry in a region, and in accordance with industry growth rate; it should increase the amount of the whole nation. The industrial structure component refers to the industry growth rate and the state’s overall growth rate difference caused by regional industry growth. The competitiveness component refers to the actual growth differences of regional industry changes. Respectively, these components indicate the effect of the nation’s growth, industrial structure, and competitiveness. However, this approach deviates from the traditional analysis method, which is mainly used for industrial competitiveness or industrial structure analysis, even if combined with industry shift, it does not alter the traditional analysis method, such as the extension of traditional dynamic deviation—the share analysis model and quantitative analysis of industrial transformation of the regional units—this model gives full consideration to the national industry increment problem from the change in industrial quantity in a single region to the measure of the change in industrial quantity across regions (Formula (2)). From theoretical analysis and reality, the sum of all the regional industry transfer components should be 0. Combined with the traditional deviation analysis (Model (1)), the remainder of regional industry growth, after removing national growth components and sector structure, is the measure of industry transfer, and it is an area owing to its reduction that arises from the difference and the average development that has increased.
(1)$$\Delta X_{ij}=X_{ij}^{\prime }-X_{ij}=X_{ij}r+X_{ij}(r_{i}-r)+X_{ij}(r_{ij}-r_{i})$$

Among them, $r=\sum_{i=1}^{S}\sum_{j=1}^{R}\left(X_{ij}^{\prime }-X_{ij}\right)/\sum_{i=1}^{S}\sum_{j=1}^{R}X_{ij}$; $r_{i}=\sum_{j=1}^{R}\left(X_{ij}^{\prime }-X_{ij}\right)/\sum_{j=1}^{R}X_{ij}$
$$r_{ij}=\left(X_{ij}^{\prime }-X_{ij}\right)/X_{ij}$$
(2)$$\begin{array}{l} \sum_{j=1}^{R}\left(X_{ij}^{\prime }-X_{ij}\right)=[X_{i1}r+X_{i1}(r_{i}-r)+X_{i1}(r_{i1}-r_{i})]+[X_{i2}r+X_{i2}(r_{i}-r)+\\ X_{i2}(r_{i2}-r_{i})]+\cdots +[X_{iR}r+X_{iR}(r_{i}-r)+X_{iR}(r_{iR}-r_{i})] \end{array}$$
(3)$$\sum_{j=1}^{R}\left(X_{ij}^{\prime }-X_{ij}\right)=\sum_{j=1}^{R}\Delta X_{ij}=\sum_{j=1}^{R}X_{ij}r+\sum_{j=1}^{R}X_{ij}(r_{i}-r)+\sum_{j=1}^{R}X_{ij}(r_{ij}-r_{i})$$

Among them, $r=\sum_{i=1}^{S}\sum_{j=1}^{R}\left(X_{ij}^{\prime }-X_{ij}\right)/\sum_{i=1}^{S}\sum_{j=1}^{R}X_{ij}$; $r_{i}=\sum_{j=1}^{R}\left(X_{ij}^{\prime }-X_{ij}\right)/\sum_{j=1}^{R}X_{ij}$
$$r_{ij}=\left(X_{ij}^{\prime }-X_{ij}\right)/X_{ij}$$
(4)$$CRIT=\sum_{j=1}^{R}X_{ij}(r_{ij}-r_{i})=X_{i1}(r_{i1}-r_{i})+X_{i2}(r_{i2}-r_{i})+\cdots +X_{iR}(r_{iR}-r_{i})$$

The industrial added value of 31 provinces and cities is selected here for measurement. The inter-regional industrial transfer is mainly manufacturing, so the agricultural, construction, and service industries are not considered here. The industrial added value comes from the *China Statistical Yearbook* and the statistical yearbook of provinces and cities over the years. In this study, the measurement results are verified again to ensure that the sum of industrial transfer increment and industrial transfer reduction is 0. The datum of the current year is the final variable value, and the industrial added value of the previous year is the initial variable value. The numerical results are presented in the Table 2.

### 3.3. Measurement of Ecological Efficiency

KLEM model was applied for the input–output relationship of water resources utilization in the process of economic growth, which decomposed input into labor, capital, energy, and intermediate input. Output refers to the desirable production of economic significance. In the existing measurement studies of energy efficiency or carbon emission efficiency, only total energy consumption and total carbon dioxide emission are included in the model, while in the existing measurement studies of water efficiency, only total water consumption and total wastewater discharge are included in the measurement model. Another important theory concerning ecological efficiency is the hypothesis of the environmental Kuznets curve (refer to [39,40]) indicating that, after suffering from a terrible period of low ecological efficiency, situations would improve by further development of the economy.

Concerning the problems of ecological efficiency in the development of the Yangtze River economic belt, this paper considers ecological efficiency as it relates to water resources and water environmental problems. Additionally, energy consumption and air pollution problems are included. Capital, labor, energy, and water serve as the input indicators, while GDP, wastewater discharge, sulfur dioxide emissions, and solid waste are the designated outputs. Because of data collection problems, other air pollution indicators are not applicable. The selection of major indicators is as follows.

➀Capital investment. Research by [41] on calculating the perpetual inventory method, based on new investment and considering the economic depreciation of the capital stock in the previous years, sets the depreciation rate as 9.6%, owing to technical innovation and capital waste. This begins in 1992 for the accounting year used here. The formula for its calculation is $K_{it}=K_{i,t-1}^{*}(1-\delta_{it})+I_{it}$, where $K_{it}$ and $K_{i,t-1}^{*}$ are capital stock for the current year and final year, $\delta_{it}$ is the economic depreciation rate of the year, and the capital stock of each province from 2005 to 2018 is calculated based on data from the *China Statistical Yearbook*.

➁Labor input. This article uses the total number of workers (urban and rural) at the end of the year in the provinces to represent the regional labor input. Owing to current university graduates, employment is difficult and the education level is also rising. This paper does not consider labor, human capital problems in past years, and related basic data on population and employment statistics that are available in the *China Statistical Yearbook*. Missing data for some years in individual provinces were calculated from other years according to the smoothing index, and some data were also obtained from provincial statistical bulletins.➂Total amount of water used in the region. Regional economic growth studies generally choose total water consumption as the input index of water resources, including agricultural, industrial, domestic, and ecological water. The analysis here is not specific to particular industries, and domestic water and ecological water are also considered as indispensable links of economic life. Relevant data come from the *China Statistical Yearbook* and *China Water Resources Bulletin.*➃Total energy consumption. The annual consumption of various types of energy in each province is converted into the total amount of standard coal for measurement. Natural gas, coal, oil, hydropower, etc., in energy consumption are not considered separately, and production energy consumption and domestic energy consumption are no longer distinguished. Relevant data come from the *China Statistical Yearbook* and the *China Energy Statistical Yearbook*, among others. In some years, some data also come from provincial statistical bulletins.➄Desired output indicator—real GDP. Excluding the influence of price changes in the study, the actual GDP of each province is chosen to represent the desired output, and the calculation method is $GDP_{it}^{R}=GDP_{it}^{N}\times I_{it}$, where $GDP_{it}^{N}$ is the nominal GDP of the province in 2003, $I_{it}$ is the growth index relative to the base period of 2003, and $GDP_{it}^{R}$ is the actual GDP denominated in the base period of 2003, which is calculated according to the *China Statistical Yearbook* for past years.

➅Nondesirable output index—3 sorts of waste discharge. Total wastewater discharge, total sulfur dioxide discharge, and industrial solid waste production are selected here to represent the release of the three wastes. These represent three specific types of environmental pollution, including water, air, and land pollution. Relevant data come from the *China Statistical Yearbook* and the *China Environmental Statistical Yearbook* over the years; part of the data for the three wastes also comes from the statistical yearbook of provinces and cities. Total waste gas discharge is not selected because data are unavailable

Here, the se-SBM model considering nondesirable output (discharge of three wastes) is used to measure the ecological efficiency of the Yangtze River economic belt. The research period is from 2005 to 2018. To reflect the dynamic comparison of ecological efficiency measurement, the selection window period is 14 years. The weights of wastewater discharge, sulfur dioxide discharge, and solid waste are set as 1/3. Accordingly, input-oriented ecological efficiency measurement results of the Yangtze River economic belt can be obtained (as shown by Table 3).

## 4. Model of Industrial Transfer Driven by Environmental Regulation

In this section, we reconstruct the dynamic model of industrial transfer from 2005 to 2018 for each region of the Yangtze River economic belt by using the Markov process as the basic framework of the model. The driving mechanism of environmental regulation on industrial transfer is reflected in the parameter setting of the Markov transfer probability matrix. Owing to the introduction of the influencing factors of environmental regulation, the Monte Carlo simulation of industrial transfer encounters a series of mathematical and technical difficulties (such as the need to prove the existence of a model that satisfies the assumptions) that preclude achieving a reasonable and accurate numerical simulation.

### 4.1. Establishment of the Markov Model of Industrial Transfer

To study the correlation between the time series of HJ’s green technology innovation investment and the time series of carbon performance in the most recent five years, and based on the numerical results of carbon performance calculated in the previous section, this section first needs to obtain the time series of HJ’s green technology innovation investment in the most recent five years.

To describe the dynamics of industrial transfer, the cumulative amount of industrial transfer between two provinces and cities from 2005 to 2018 is used as a set of significant statistics. However, we can only collect the final transfer volume of industries in each province and city in each year (a positive value indicates that all transfers in and out are a net transfer in, and a negative value indicates a net transfer out). Based on this, it is impossible to obtain data for every two provinces and cities for each year, which is the amount of industrial transfer between them. To this end, we use the Monte Carlo simulation method to simulate the industrial transfer values between provinces and cities for each year, based on the existing increase and decrease in industrial transfer in each province and city and the geographical information for each province and city, forming 14 (2005–2018) values. These are used to create the industrial transfer matrix between provinces and cities for each year.

As for the industrial transfer volume for each province and city in each year, we have already prepared the data, which is presented in Table 2 and shows the increase and decrease in industrial transfer in 11 provinces and cities of the Yangtze River economic belt and other regions during 2005 to 2018.

Behind the macroprocess of industrial transfer is the microbehavior of the cross-regional transfer of specific enterprises and industries in each region. The average probability of transferring enterprises or industries from province and city i to province and city j in the Yangtze River economic belt in a certain year t can be recorded as T(i,j,t); then, for each fixed year t, T(i,j,t): i∈{1, 2, …, 12} constitutes a transition probability matrix, where {1, 2, …, 12} represents the 12 regions.

Since each value in Table 2 is the increase or decrease in the industry in the region in that year, a negative value indicates that the industry’s transfer-out is greater than the transfer-in, and a positive value indicates that the industry’s transfer-in is greater than the transfer-out. If this cannot be obtained, then O(i,j,t) denotes the amount of industrial transfer from area i to area j during year t, and I(i,j,t) is the industrial transfer from area i from area j during year t. The increase or decrease in the industrial transfer in the region i in this year is then I(i,j,t)-O(i,j,t) fixed (i,t), when j is taken over {1, 2, …, 12}. Denote O(i,t) as the total industrial transfer out of area i in the year t, that is, O(i,j,t) fixed (i,t). When j takes the sum of the items of {1, 2, …, 12}, then record I(i, t) is the total industrial transfer in area i during year t, that is, I (i,j,t) fixed (i,t), when i takes the sum of the terms of {1, 2, …, 12}, thus, I(i,t)-O(i,t). That is, the increase or decrease in industrial transfers in region i during year t. The transition probability matrix T constructed above can be used to build the relationship between O(i,j,t) and O(i,t), given the known values of O(i,t) and O(i,j,t). For each value of j, that is, for the distribution of O(i,t) during year t in region i among regions, O(i,t) is multiplied by the probability of transferring to region j T(i,j,t) to obtain an estimate of O(i,j,t).

For the construction and estimation of the transition probability matrix T, this paper mainly considers the influence of two aspects, one is the active role of environmental regulation on the transition probability, and the other is the geographical information between regions. According to the first law of geography, everything is spatially related, and items that are close to each other are more spatially related than items that are far apart. By default, the amount of industrial transfer between adjacent areas is more probable than nonadjacent ones. There is a large amount of industrial transfer between regions. Therefore, this paper uses the geographic information matrix, that is, the adjacent matrix {d(i,j)} between provinces and cities in the Yangtze River economic belt, and the element d(i,j) in the matrix expresses region i and region j. If it is adjacent, then it is recorded as 1, and it is recorded as 0 if it is not adjacent. For the distance between any area of the Yangtze River economic belt and the outside of the Yangtze River economic belt, it is considered to be weakly adjacent, and it is recorded as 0.1. The information of environmental regulation is reflected in Table 1, and the data in this table are recorded as a matrix A of the form n × m, where n = 12 represents 12 areas and m = 14 represents 14 years, and the last column is the intensity of environmental regulation outside the Yangtze River delta, and its value is taken from the mean. This section characterizes the relationship between environmental regulation and the transition probability matrix in the following expression: $T=C+k_{1}A+k_{2}\mathrm{Ad}$. Among the variables, C is the coefficient matrix, which reflects the influence of other variables on environmental regulation and transition probability, and k_1_ and k_2_ are coefficients. A × d constitutes the intersection term of environmental regulation and geographic information, thus constituting a nonlinear regression model. However, given the above description, the transition probability matrix and O(i,j,t), O(i,t) relationship, and O(i,j,t), O(i,t) and the quantitative relationship between the increase and decrease in the actual industry in each region in each year is constrained by matrix A(:,:,t) (that is, the transition probability matrix in the t-th year). With O(:,t) representing the vector {O(1,t), O(2,t), …, O(n,t)} and I(:,t) representing the vector {I(1,t)), I(2,t), …, I(n,t)}, then N(:,t) represents the vector {N(1,t), N(2,t), …, N(n,t)}, where N(i,t) represents the net value of the actual industrial increase or decrease in region i during year t. Then, the following matrix relationship is obtained:(5)O(:,t)×T(:,:,t)=I(:,t)
(6)N(:,t)=I(:,t)−O(:,t)
based on this matrix equation, O(:,t) × T(:,:,t) = N(:,t) + O(:,t), O(:,t) × (T(:,:, t) − E) = N(:,t), and O(:,t) = N(:,t) × inv(T(:,:,t) − E) and I(:,t) = N(:,t) + O(:,t) = N(:,t) × (inv(T(:,:,t) − E) + E), where E is the identity matrix, and inv(T) represents the inverse matrix of matrix T. Since all elements of O(:,t) and I(:,t) are positive, the following constraints are obtained:(7)N(:,t)×inv(T(:,:,t)−E)>0
(8)N(:,t)×(inv(T(:,:,t)−E)+E)>0

In addition, T(:,:,t) as a transition probability matrix should satisfy the following two conditions:(9)T(i,j,t)∈[0,1] and T(i,i,t)=0 holds for any i,j∈{1,…,n}
(10)$$T\left(i,i,t\right)\times {\left(1,\ldots ,1\right)}^{\prime }={\left(1,\ldots ,1\right)}^{\prime }$$

Therefore, in this expression of the transition probability matrix constructed with $T=C+k_{1}A+k_{2}\mathrm{Ad}$, the assignments of C, k_1_, and k_2_ are limited, and wrong assignments will not satisfy these two equality constraints, so further Monte Carlo simulation of the model will encounter mathematical difficulties: the first item to demonstrate is that, given any vector N(:,t) of the form 1 × n, whether there is a condition that satisfies the above constraints (7) and (8), and matrix T(:,:, t) of its own conditions (9) and (10) exists, whether it exists uniquely or the linear space of solution has a large degree of freedom. To this end, this analysis completes the argument with the following lemma. In fact, this lemma can be accepted intuitively: by randomly generating a large number (100,000 times) of matrices that satisfy the conditions (9) and (10), none of them satisfy conditions (7) and (8); therefore, in order to achieve mathematical rigor, the approach presented in this paper must eliminate the question of the existence of the T matrix, satisfying the conditions through the lemma in the next section.

### 4.2. Matrix Analysis of the Markov Model of Industrial Transfer

To quantitatively analyze the enhancement and attenuation law of the impact of HJ’s green technology innovation investment on carbon performance over time, this subsection mainly studies the term structure of the impact of green technology innovation investment on carbon performance, that is, the law of the impact of green technology innovation investment on carbon performance over time. For the measurement of impact degree, this section calculates the impact degree based on the correlation analysis of time series.

To prove the existence of the matrix in the Markov model of industrial transfer established in Section 4.1, this section first considers the following lemma and proves it.

**Lemma** **1.**
*For any vector N of the form 1 × n, there exists a matrix T of the form n × n that satisfies (7)–(10), and T that satisfies the condition is not unique.*


**Proof:** On the premise that t remains unchanged, the vectors I(:,t), O(:,t), and N(:,t) in the form of 1 × n are abbreviated as I_n, O_n, N_n. The equation that communicates the relationship between O_n, I_n, and T is:
(11)$$O_{n}\times T=I_{n}$$The equation that reflects the connection between I_n, O_n, and N_n is:
(12)$$N_{n}=I_{n}-O_{n}$$Based on Equation (11), the following two matrix equations are obtained:
(13)$$\left(1,\cdots ,1\right)\cdot \left(\begin{array}{ccc} O\left(1\right) & 0 & 0\\ 0 & \ddots & 0\\ 0 & 0 & O\left(n\right) \end{array}\right)\cdot T=\left(I\left(1\right),\cdots ,I\left(n\right)\right)$$
(14)$$\left(1,\cdots ,1\right)\cdot T^{\prime }\cdot \left(\begin{array}{ccc} O\left(1\right) & 0 & 0\\ 0 & \ddots & 0\\ 0 & 0 & O\left(n\right) \end{array}\right)=\left(O\left(1\right),\cdots ,O\left(n\right)\right)$$Observing the two formulas, (13) and (14), it is found that the matrix should be defined as
(15)$$M:=\left(\begin{array}{ccc} O\left(1\right) & 0 & 0\\ 0 & \ddots & 0\\ 0 & 0 & O\left(n\right) \end{array}\right)\cdot T$$Thus, Formulas (13) and (14) become:
(16)$$\left(1,\cdots ,1\right)\times M=I_{n}$$
(17)$$\left(1,\cdots ,1\right)\times M^{\prime }=O_{n}$$Subtract the two Equations (16) and (17), to obtain
(18)$$\left(1,\cdots ,1\right)\times D=N_{n}$$Equation (12) is used here, where matrix D is defined as follows:
(19)$$D=M-M^{\prime }$$Obviously, matrix D is a symmetric matrix. Since the diagonal elements of matrix T are all 0, the diagonal elements of matrix M are also all 0, so the diagonal elements of matrix D are also all 0, and the elements of D can be positive or negative. So far, the existence of the solution of the original problem has been reduced to the existence of the solution of Equation (18). By transposing, Equation (18) is equivalent to D × (1, …, 1)’ = N_n’, that is, a system of equations composed of n first-order equations, since matrix D has all 0 diagonal elements, then it is a symmetric matrix of order n, thus having n(n−1)/2 variable elements. Therefore, there are at least n(n − 1)/2 − n or n(n − 3)/2 solutions for the system of Equation (18). It can be verified that when n = 3, the system of Equation (18) may have no solution. However, in this analysis, n = 14, so there are at least 77 sets of solution vectors. It can be seen that there is still a large degree of freedom when stochastic simulation of matrix D is performed.After matrix D is obtained, matrix M can be further randomly determined, but it is necessary to ensure that each element of matrix M is non-negative and the diagonal elements are all 0. It can be realized as follows: a real number r between 0 and 1 can be set as the import and export proportional coefficient. For the element d(i,j) in matrix D, if it is 0, then let m(i,j) = m (i,j), which can be any random positive number, the value of the same order of magnitude as N_n is suitable; if it is a positive number, then let m(i,j) = d(i,j) × (1 + r), m(j,i) = d(i,j) × r, thus satisfying the condition; if the element d(i,j) is negative, then let m(i,j) = −d(i,j) × r,m(j,i) = −d(i,j) × (1 + r); the condition can still be satisfied.After matrix M is obtained, use the Formula (15) to perform linear row transformation, that is, multiply each row of M by 1/O(i), so that the sum of the rows of the resulting matrix T is 1, as long as each O(i). The value is the row sum of the i-th row of M. Note that each row of matrix M is positive except for the elements on the diagonal, so the row sum is not 0, thus avoiding O(i) being 0 or a negative value. Thus, matrix T has been determined.It can be seen from this construction and discussion that a matrix T of the form n × n that satisfies the conditions (7)–(10) exists and is not unique, and matrix T satisfying (7)–(10) can be determined by a linear space with at least n(n − 3)/2 degrees of freedom. Therefore, it is feasible to introduce a random fitting of the form $T=C+k_{1}A+k_{2}\mathrm{Ad}$ to matrix T and perform Monte Carlo simulation.□

## 5. Numerical Implementation of Industrial Transfer Driven by Environmental Regulation

Section 4 constructs the model of dynamic simulation of industrial transfer driven by environmental regulation. On this basis, this section designs and implements the numerical simulation of the dynamic model of industrial transfer driven by environmental regulation.

### 5.1. Numerical Simulation of the Dynamic Model of Industrial Transfer

In Listing 1, the simulation first designs and generates matrix D with the least restrictive conditions, as specified in Section 5 (i.e., M1 in the Listing 1 below), and reflects the proportional relationship between elements determined by $T=C+k_{1}A+k_{2}\mathrm{Ad}$ in the process of random assignment. It also satisfies that D is a symmetric matrix and ensures that the matrix represented by Equation (18) holds. The algorithm embodied in the code gradually realizes the transposed form of Equation (18) for matrix D from the upper left corner to the lower right corner: D × (1, …, 1)’ = N_n’, because the D matrix is symmetric, it is determined. When the elements of a certain row are selected, the corresponding elements of a certain column are also determined at the same time, but this specializes the order of each region in the matrix, and the first region reflects the ratio between the elements defined by $T=C+k_{1}A+k_{2}\mathrm{Ad}$. To eliminate the deviation caused by this specialization, the algorithm designs a random wrapping mechanism, and generates a random sequence through the function randperm that comes with MATLAB, which is recorded as in1, and the recovery sequence obtained by calculating in1 is recorded as in2, so that at the beginning of generating the random matrix, we wrap the line with the in1 sequence, and then use the sequence in2 for column transformation and row transformation to restore the original order after generation, that is, in the code: M1 = M1(in2,:); M1 = M1(:,in2). In addition, in the process of determining the elements of matrix D, in order to make the sum of each column meet the requirements, the last three columns cannot be set randomly, and the three undetermined elements are determined by solving the cubic equation with 3 unknowns.

After the D matrix is obtained, we follow the steps described in Section 4.1 to set a real number r between 0 and 1 as the import and export proportional coefficient. For the element d(i,j) in matrix D, if it is 0, then let m(i,j) = m(i,j), any random positive number can be used, the value of the same order of magnitude as N_n is suitable; if it is a positive number, then let m(i,j) = d (i,j) × (1 + r), m(j,i) = d(i,j) × r, thus satisfying the condition; if the element d(i,j) is negative, let m(i,j)) = −d(i,j) × r,m(j,i) = −d(i,j) × (1 + r), the condition can still be satisfied. Thus, the M matrix is obtained. After matrix M is obtained, we use Formula (15) to perform linear row transformation, that is, multiply each row of M by 1/O(i), so that the sum of each row of the resulting matrix T is 1, then let each O(i) be the sum of the i-th row, and whose value is M, which is sufficient.

There are two coefficients in the code that can be adjusted. One is the distance influence intensity coefficient, which is represented as “pod” in the code, and its value can be 0.8, 0.5, and 0.2, indicating that the distance influence intensity factor has a strong, normal, or weak effect on industrial transfer, respectively. The second is the average coefficient of import and export ratio, mentioned above, which is represented by “oirate” in the code, and its value is set to 0.2; other values can be considered, such as 0.1 or 0.5. This is shown specifically in Listing 1.

**Listing 1.** Codes to simulate the cumulative directional transfer of industries.

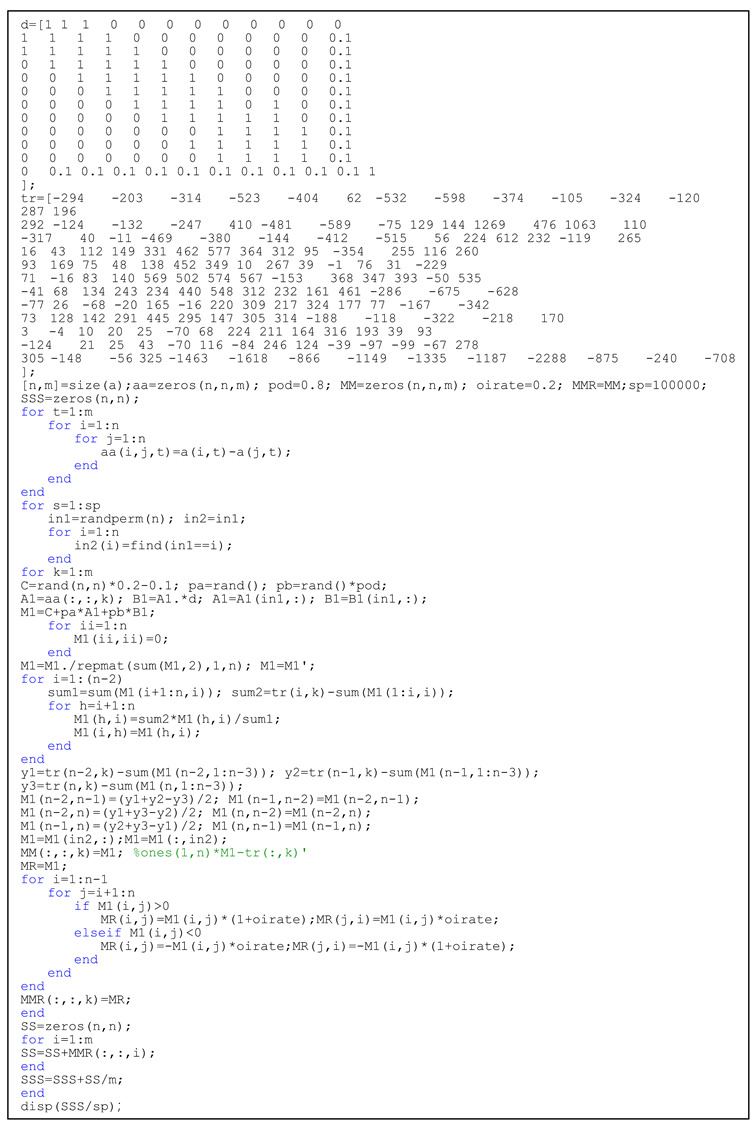



### 5.2. Analysis of Numerical Results of the Dynamic Model of Industrial Transfer

Table 4 provides values of the cumulative directional transfer of industries in the 11 provinces and cities in the Yangtze River economic belt, directly generated by the code given above. Note that the result is generated by Monte Carlo simulation, and each run receives a copy, as shown in Table 4. There are differences in the value of the estimated results, but the relative relationship between the data is roughly the same.

The distance influence intensity coefficient in the model has an impact on the final estimation result. Therefore, the analysis used in this paper calculates the cumulative transfer volume of the industry when the distance influence intensity coefficient is 0.5, 0.8, and 0.2, according to the situation. The results are shown in Figure 1, Figure 2 and Figure 3; the average in–out ratio coefficient is set to 0.2 in each case.

Figure 1 shows that when the distance influence intensity coefficient is set to 0.5, the cumulative value of industrial transfer between regions in the Yangtze River economic belt reflects the influence of the relative strength of environmental regulation and whether the regions are adjacent. According to the environmental regulation intensity index of the 11 provinces and cities in the Yangtze River economic belt shown in Table 1 (Section 3), it can be seen that the environmental intensity of the 14 years from 2005 to 2018 varies among regions, among which Sichuan, Guizhou, and Hubei have greater environmental regulation intensity. As shown in Figure 1, Hubei’s industrial transfer-out volume to Shanghai, Jiangsu, Zhejiang, and Anhui is relatively high; Sichuan and Guizhou have higher industrial transfer-out volume than Yunnan, Chongqing, Jiangxi and other provinces and cities. This shows that the model’s characteristics of the influence of environmental regulation on industrial transfer in the Yangtze River economic belt have been portrayed and reflected. The influence of geographical factors on the amount of industrial transfer between regions in the Yangtze River economic belt also has a certain performance in the quantitative comparison shown in Figure 1: if Jiangsu, Zhejiang and Shanghai are adjacent, the industrial transfer between these three provinces and cities is comparable. Note that there is always a vacancy in each cluster in the figure, and that vacancy is the amount of industry transfer from region to region, set to 0. Through observation, it is found that the number of bars near each space is relatively high, which means that areas near each other received relatively more industrial transfers.

In the case of increasing the distance influence strength coefficient to 0.8, the histogram of the cumulative directional transfer amount in each area is shown in Figure 2. Compared with Figure 1, the most obvious difference is that Figure 2 shows that there are greater fluctuations in the amount of industrial transfer from different regions. There are only two in Figure 1. At the same time, in Figure 2, there are many more inter-regional industries whose one-way transfers are less than 2000, and some are even less than 1000. Obviously, this change is affected by the upward adjustment of the distance influence intensity coefficient. The maximum transfer amount visible in Figure 2 is the transfer out of Jiangsu to Shanghai. At the same time, the transfer out of Shanghai to Jiangsu is also close to 12,000. In comparison, the difference between Shanghai and Jiangsu is not large, and the large amount of industrial transfer results from the fact that the absolute number of industries in the two places is relatively large, and that the two places are adjacent to each other and have convenient transportation. In Figure 2, the smallest amount of industrial transfer out is from Hunan to Yunnan and Yunnan to Anhui. In fact, Hunan and Yunnan are separated by a province, and the distance is not particularly far, but the geographic information matrix in this model only reflects whether or not it is adjacent, so the final estimated amount of industrial transfer is still small. In addition, it can be seen from Figure 2 that the amount of industrial transfer from Jiangsu and Zhejiang to the west is relatively small. On the one hand, owing to the fact that the two provinces are not adjacent, more industries from Jiangsu and Zhejiang are allocated to neighboring provinces and cities. On the other hand, it results from the high coefficient of environmental regulation intensity in Guizhou, western Sichuan, and other places.

In the case where the distance influence intensity coefficient is adjusted down to 0.2, the histogram of the cumulative directional transfer amount in each area shown in Figure 3 is obtained. Comparing with Figure 2, it is found that the most obvious difference is that the fluctuation of the industrial transfer volume in each region shown in Figure 3 is significantly smaller. In Figure 3, there is no industry transfer from one province or city to another province or city close to 12,000, while the industrial transfer volume between all provinces and cities exceeds 1000. The geographical connection of the industrial transfer out of all regions is significantly weakened, and the industrial transfer out and in-transfer volume outside the Yangtze River economic belt have increased significantly. However, after the coefficient of the geoinformation matrix is reduced, the effect of environmental regulation intensity on inter-regional industrial transfer becomes obvious. As mentioned above, Hubei has a relatively high intensity of environmental regulation. In Figure 2, Hubei becomes the province and city with the largest amount of industrial transfer.

From the analysis of the characteristics of these three figures, it can be seen that the dynamic model of industrial transfer in the Yangtze River economic belt constructed in Section 4 and the numerical simulation demonstrated in this section can reflect the design intention of the model under the discussion of different scenarios where distance affects different intensities, which may be referred to as the law of influence on industrial transfer. In the sequel, we construct the relationship between the cumulative industrial transfer out and ecological efficiency, so as to further examine the effect of environmental regulation intensity on environmental efficiency by driving industrial transfer.

## 6. Verification of the Eco-Efficiency Driving Mechanism of Environmental Regulation

To establish the numerical correlation between the dynamics of industrial transfer and ecological efficiency, based on the industrial transfer volume between regions of the Yangtze River economic belt obtained in Section 5, the differences of eco-efficiency and industrial transfer as well as environmental regulation intensities between pairs of provinces and cities will be considered. Table 5 shows the cumulative directional transfer of industries in the 11 provinces and cities in the Yangtze River economic belt (the distance influence intensity coefficient is set to 0.5, and the average entry and exit ratio coefficient is set to 0.2).

We consider other control variables: economic development level, technological innovation level, external dependence, urbanization level, industrial structure, per capita education level, level of opening to the outside world, per capita water resources endowment, and energy structure. Accordingly, the regression analysis is still carried out with the difference model:(20)$${\Delta E}_{c}=k_{0}{\Delta T}_{r}+k_{1}{\Delta X}_{1}+k_{2}{\Delta X}_{2}+k_{3}{\Delta X}_{3}+k_{4}{\Delta X}_{4}+k_{5}{\Delta X}_{5}+k_{6}{\Delta X}_{6}+k_{7}{\Delta X}_{7}+k_{8}{\Delta X}_{8}+k_{9}{\Delta X}_{9}+\varepsilon$$

Among them, ε is the random error term and Tr represents the proportion of industrial transfer to local GDP; Δ represents the variable difference between two regions, such as ΔT_r_ , which is no longer a column vector of 11 values, but a column vector with 55 values (i.e., 11 × (11−1)/2), which lists all the values in the upper half of the matrix in Table 2: {9211, 9862, …, 7170, 6613, …, 3562, 7721, …, 1352}.

In addition, when using the data in Table 1, we divide it by the 14-year average GDP for each region, and the order of regions in the table follows: 279,880.94, 726,910.95, 463,102.45, 226,151.21, 170,745.42, 294,842.58, 288,557.36, 153,615.71, 311,049.93, 99,812.41, 138,745.42, 294,842.58, 288,557.36, 153,615.71, 311,049.93, and 99,812.41. This is the ratio of export volume to local GDP. Listing 2 shows the implementation of all the steps:

**Listing 2.** Codes of regression between industrial transfer and ecological efficiency.

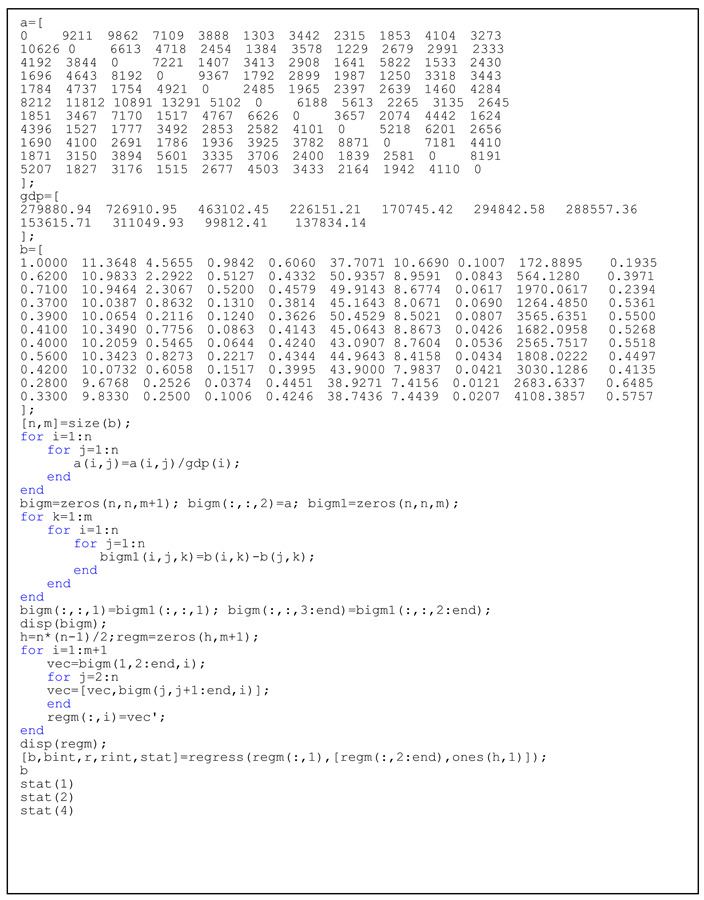



The following regression expression is obtained after executing the program:(21)$${\Delta E}_{c}=-0.0672{\Delta T}_{r}+0.1121{\Delta X}_{1}-0.2062{\Delta X}_{2}+0.5714{\Delta X}_{3}+1.8610{\Delta X}_{4}-0.0006{\Delta X}_{5}+0.0075{\Delta X}_{6}+1.5840{\Delta X}_{7}+0.0000{\Delta X}_{8}-0.6932{\Delta X}_{9}+0.0062$$
where stat(1) = 0.9954, stat(2) = 943.8971, and stat(4) = 3.07 × 10^−4^. That is, the fitting degree of the regression model is 0.9954, the F value is 943.8971, and the error sum of squares is 3.07 × 10^−4^. It can be seen from the results that the fitting degree of the regression model is 0.9954, which is a very high degree of fitting; the F value is 943.8971, which is also very high, again indicating that the regression fitting is significantly credible. It is within the acceptable error range. Therefore, the regression results show that the cumulative amounts of industrial transfer between each pair of provinces and cities in the 14 years from 2005 to 2018 have significant correlations; it affected the gaps of the ecological efficiency between the two provinces and cities in 2018, thus indicating industrial transfer does change the eco-efficiency of provinces and cities in the Yangtze River economic belt.

As far as the regression coefficient is concerned, the coefficient of ΔT_r_ is negative, indicating that the transfer of industries between provinces and cities promotes the improvement of ecological efficiency. Under the goal of minimizing the sum of squared errors, we performed regression tests on the defaults for different variables in the model multiple times, and executed Listing 3 on the basis of Listing 2.

We ran the results of Listing 3 to obtain the expression for the regression fit:(22)$${\Delta E}_{c}=-0.0222{\Delta T}_{r}+0.1167{\Delta X}_{1}+0.5268{\Delta X}_{3}+2.0207{\Delta X}_{4}+1.7691{\Delta X}_{7}+0.0000{\Delta X}_{8}-0.6899{\Delta X}_{9}+0.0044$$
**Listing 3.** Codes of regression of industrial transfer and ecological efficiency.
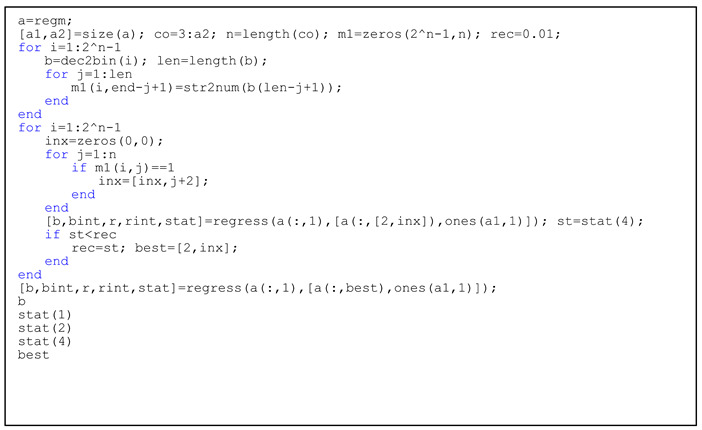

where stat(1) = 0.9953, stat(2) = 1213.9, stat(4) = 2.9854 × 10^−4^. That is, the fitting degree of the regression model is 0.9853, the F value is 1213.9, and the error sum of squares is approximately 0. The result shows that the regression fitting result is acceptable. The following are significant control variables: ΔX_1_ economic development level, ΔX_2_ scientific and technological innovation level, ΔX_3_ external dependence, ΔX_4_ urbanization level, ΔX_7_ level of opening to the outside world, and ΔX_9_ energy structure. Other control variables have little effect on the mechanism of environmental regulation affecting ecological efficiency.

Table 6 shows the cumulative directional transfer of industries in the 11 provinces and cities in the Yangtze River economic belt (the distance influence intensity coefficient is set to 0.8, and the average in–out ratio coefficient is set to 0.2).

We consider other control variables: economic development level, technological innovation level, external dependence, urbanization level, industrial structure, per capita education level, level of opening to the outside world, per capita water resources endowment, and energy structure. Accordingly, the regression analysis is still carried out with the difference model:(23)$${\Delta E}_{c}=k_{0}{\Delta T}_{r}+k_{1}{\Delta X}_{1}+k_{2}{\Delta X}_{2}+k_{3}{\Delta X}_{3}+k_{4}{\Delta X}_{4}+k_{5}{\Delta X}_{5}+k_{6}{\Delta X}_{6}+k_{7}{\Delta X}_{7}+k_{8}{\Delta X}_{8}+k_{9}{\Delta X}_{9}+\varepsilon$$

Among them, ε is the random error term, Tr represents the proportion of industrial transfer to local GDP, and Δ represents the variable difference between two regions. Similar to the case above for the distance effect strength factor of 0.5, running the same code yields:(24)$${\Delta E}_{c}=0.1838{\Delta T}_{r}+0.1218{\Delta X}_{1}-0.2001{\Delta X}_{2}+0.5423{\Delta X}_{3}+1.8727X_{4}-0.0005{\Delta X}_{5}-0.003{\Delta X}_{6}+1.6722{\Delta X}_{7}+0.0000{\Delta X}_{8}-0.6857{\Delta X}_{9}-0.0004$$
where stat(1) = 0.9954, stat(2) = 952.0613, stat(4) = 3.0454 × 10^−4^. That is, the fitting degree of the regression model is 0.9954, the F value is 952.0613, and the error sum of squares is 3.0454 × 10^−4^. It can be seen from the results above that the fitting degree of the regression model is 0.9954, and the fitting degree is very high; the F value is 952.0613, which is very high, again indicating that the regression fitting is significantly credible; the error sum of squares is 3.0454 × 10^−4^ (i.e., 0.000307), which is also within the acceptable error range. Therefore, the regression equation above is significantly established.

However, note that the regression coefficient 0.1838 of the ΔT_r_ term is positive, which is opposite to the sign of the regression coefficient of the ΔT_r_ term above for the case where the distance influence strength coefficient is 0.5. This is due to the increase in the distance influence intensity coefficient to 0.8 (owing to the fact that the determination of the distance influence intensity coefficient cannot be supported by empirical evidence; the quantitative verification process below is verified separately in the three cases of strong, medium, and weak inter-provincial distance influence), so that geographical information becomes the dominant factor in determining the ratio of transition probability among variables, making it difficult to reflect differences in environmental regulation. Therefore, the actual meaning of the positive regression coefficient of the ΔT_r_ term is not credible. Similarly, when the distance influence intensity coefficient is 0.2, the influence of environmental regulation is prominent. We repeat this process to obtain:(25)$${\Delta E}_{c}=-0.0392{\Delta T}_{r}+0.0173{\Delta X}_{1}-0.7141{\Delta X}_{2}+0.9170{\Delta X}_{3}+0.0781{\Delta X}_{4}+4.3131{\Delta X}_{7}+0.0000{\Delta X}_{8}-1.8003{\Delta X}_{9}+0.0312$$

This result is consistent with the analysis given above. If, as predicted above, the influence of environmental regulation becomes dominant, the regression coefficient of the ΔT_r_ term returns to a negative value. In the following sections, research on environmental regulation-driven industrial transfer and eco-efficiency is discussed for the three situations.

As previously established, we the assumed a relationship between environmental regulation and the transition probability matrix: $T=C+k_{1}A+k_{2}\mathrm{Ad}$. The numerical simulation in Section 5 basically verifies the significant existence of this correlation. This section further reveals that environmental regulation is in the driving mechanism of industrial transfer on ecological efficiency. Recalling the simulation steps in Section 4: first determine matrix D from Equation (18), then determine matrix M, and then determine matrix T in Equation (15), namely,
$$\left(\begin{array}{ccc} O\left(1\right) & 0 & 0\\ 0 & \ddots & 0\\ 0 & 0 & O\left(n\right) \end{array}\right)\cdot T$$

The environmental regulation intensity matrix A is reflected in the process of simulating matrix T; considering that it cannot form a linear relationship with ΔTr, the relationship between matrix T and ΔEc is also nonlinear. Therefore, the linear relationship expression between the environmental regulation intensity matrix A and ecological efficiency cannot be obtained based on the above model. In order to verify the indirect effect of environmental regulation on ecological efficiency, and quantitatively show the relationship between the environmental regulation intensity matrix A and the ecological efficiency value, the analysis in this paper performs various forms of numerical perturbation on the environmental regulation intensity matrix A, so as to observe how the corresponding ecological efficiency changes. To this end, following the commonly used Greek value definition form of sensitivity analysis, this analysis first defines the gradient (i.e., the rate of change) Λ(ε). When the intensity matrix A is disturbed with a difference of ε, it acts on the industrial transfer matrix and eventually causes the following condition: if the magnitude of environmental efficiency is $\left|∆E_{c}^{\varepsilon }-∆E_{c}\right|$, then the value of gradient Λ(ε) is defined as $\left|∆E_{c}^{\varepsilon }-∆E_{c}\right|/\varepsilon$.

For this reason, this section considers the form of perturbing the environmental regulation intensity matrix A. Since each row of the environmental regulation intensity matrix A represents the environmental regulation intensity of different provinces and cities, the perturbation of the A matrix is carried out by row transformation. Recognizing that the matrix transformation can be divided into three, then the following three elementary transformations are used in this analysis to act on the environmental regulation intensity matrix A: line-by-line scaling transformation, two-line overall exchange, and line-by-line doubling transformation. In actual operation, each elementary transformation only changes one row (the first and third transformations) or two rows (the second); in this analysis, in order to realize that all rows are perturbed globally at one time, the same elementary transformation is carried out by multiplicative superposition (that is, decomposed into the product of multiple elementary row transformations of the same kind). The disturbance difference is represented by ε, then the first type of table exchange is defined as $\Gamma_{1}A:=M_{1}A$ (defined for any n-row matrix A), where
$$M_{1}=\left(\begin{array}{ccc} 1+\varepsilon & 0 & 0\\ 0 & \ddots & 0\\ 0 & 0 & 1+\varepsilon \end{array}\right)$$
is an n × n square matrix, the first type of transformation multiplies all elements in the A matrix by 1 + ε, and the simulation realizes the overall strengthening of environmental regulation; the second type of transformation is defined as $\Gamma_{2}A:=M_{2}A$ (for any n row of matrix A definition), where:$$M_{2}=\left(\begin{array}{cc} \begin{array}{ccc} 0 & 1 & 0\\ 1 & 0 & 0 \end{array} & \begin{array}{cc} 0 & 0\\ 0 & 0 \end{array}\\ \begin{array}{ccc} 0 & 0 & \ddots \\ 0 & 0 & 0\\ 0 & 0 & 0 \end{array} & \begin{array}{cc} 0 & 0\\ 0 & 1\\ 1 & 0 \end{array} \end{array}\right)$$
is an n × n square matrix, the second type of transformation realizes the exchange of environmental regulation strength values of two adjacent regions such as (1,2), (3,4), …, (n−1,n). If n is an odd number, the last field value does not participate in the swap. The third type of transformation is defined as $\Gamma_{3}A:=M_{3}A$ (defined for any n-row matrix A), where
$$M_{3}=\left(\begin{array}{cc} \begin{array}{ccc} 1 & 1+\varepsilon & 0\\ 0 & 1 & \;\;0 \end{array} & \;\begin{array}{cc} 0\; & 0\\ 0\; & \;\;0 \end{array}\\ \begin{array}{ccc} 0\; & \;0 & \;\ddots \\ 0\; & \;0 & \;0\\ 0\; & \;0 & \;0 \end{array} & \begin{array}{cc} 0 & 0\\ 1 & 1\\ 1+\varepsilon & 0 \end{array} \end{array}\right)$$
is an n × n square matrix, which is formed by arranging several 2 × 2 small square matrices on the diagonal, in which 1 + ε randomly appears in the lower left corner or upper right corner of the small square matrix. When 1 + ε appears in the lower left corner, then in the corresponding two rows in the A matrix, the previous row is multiplied by 1 + ε and added to the next row; when 1 + ε appears in the upper right corner, the following row is multiplied by 1 + ε and added to the previous row.

Listing 4 implements three row-transformations for rows in the environmental regulation intensity matrix A:

**Listing 4.** Codes for row-transformations of square matrix.

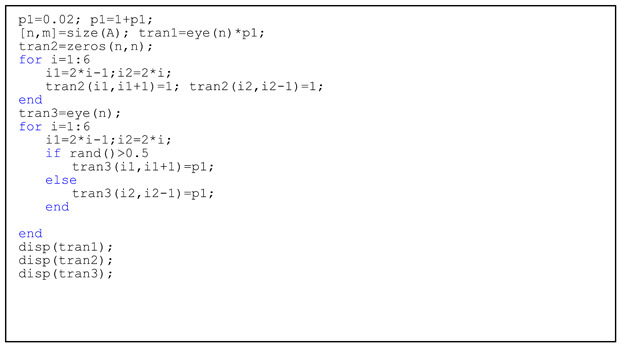



Note that the disturbance parameter set in the first and third transformations is p1 = 0.02, a positive value means that the disturbance is positive, which indicates a situation wherein environmental regulation becomes stronger, and then the following resets p1 = −0.02 to indicate that environmental regulation weakened the situation. After multiplying the three transformation matrices obtained by the codes listed above for the environmental regulation intensity matrix A (see Table 1 in Section 3), the environmental regulation intensity matrix A after three disturbances (the disturbance coefficient is 0.02) is obtained, as shown in Table 7, Table 8 and Table 9:

We next substitute the new environmental regulation intensity coefficient matrix under the three transformations presented before and repeat the steps in Section 5 to obtain the corresponding industrial cumulative directed transfer volume matrix (the distance influence intensity coefficient is set to 0.5): for Table 10, the environmental regulation intensity matrix is taken from Table 7, the distance influence intensity coefficient is set to 0.5, and the average in–out ratio coefficient is set to 0.2; For Table 11, the intensity matrix is taken from Table 8, the distance influence intensity coefficient is set to 0.5, and the average entry and exit ratio coefficient is set to 0.2. For Table 12, the intensity matrix is taken from Table 9, the distance influence intensity coefficient is set to 0.5, and the average entry and exit ratio coefficient is set to 0.2.

We reverse this regression expression to find the ΔE_c_^ε^ value in each case, and subtract it from the original ΔE_c_ value, as shown in Listing 5 (where a is the industry transfer matrix for each case):

**Listing 5.** Codes of disturbance of the three linear transformations to the environmental regulation intensity matrix.

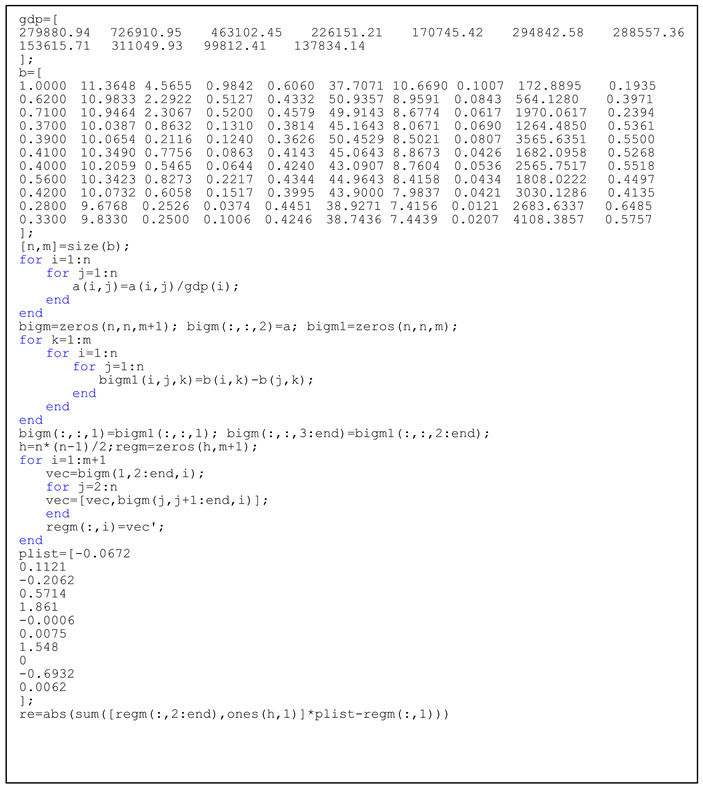



We run the code separately to obtain the difference values of 2.1109, 2.1118, and 2.1168. Based on the above definition of the gradient Λ(ε) value: |ΔE_c_^ε^ − ΔE_c_|/ε can be obtained for the disturbance of the three linear transformations to the environmental regulation intensity matrix. The gradients are 105.55, 105.59, and 105.84, respectively. Strictly speaking, the |ΔE_c_^ε^ − ΔE_c_|/ε values obtained under the second transformation cannot be called gradients because the transformation only uses line breaks and does not use the ε value. As far as the numerical results are concerned, the first transformation has the least impact on ecological efficiency, because the first transformation multiplies each value of the environmental regulation intensity matrix by 1.02, and the relative strength of the inter-regional environmental regulation intensity changes the least. The gradient obtained under the third transformation is the largest because some rows of the environmental regulation intensity matrix are numerically improved, and some rows remain unchanged, which obviously affects the relative strength of environmental regulation between regions.

Correspondingly, considering the weakening of environmental regulation, we can set p1 = −0.02 and repeat the above process and execute the code to obtain the gradients of 105.56, 105.59, and 105.79 under the disturbance of the three linear transformations to the environmental regulation intensity matrix, respectively. Obviously, the value of |ΔE_c_^ε^ − ΔE_c_|/ε obtained under the second transformation does not change, while the gradient obtained under the first and third transformations changes slightly, which can be seen in the value of the environmental regulation intensity matrix. The impact of overall scaling is not sufficiently significant, and the driving effect of environmental regulation is mainly reflected in the difference in the strength of environmental regulation between regions.

The results in this section indicate that changes in the value of environmental regulation intensity significantly affect the final value of eco-efficiency. Through this demonstration, the application constructed in this section using the regression expression of the relationship between industrial transfer and eco-efficiency shows that the realization process of environmental regulation-driven eco-efficiency improvement is indirectly promoted through the intermediate link of industrial transfer. Changes in the value of environmental regulation intensity significantly affect the final value of eco-efficiency. In the process of demonstration, the application of the regression expression of the relationship between industrial transfer and ecological efficiency constructed in this section shows that the process of environmental regulation driving the improvement of ecological efficiency is indirectly promoted through industrial transfer.

## 7. Summary and Conclusions

It can be seen that the improvement of the ecological efficiency of the Yangtze River economic belt through the path of environmental regulation significant affects or drives the transfer of industries. For carbon–green transition, we believe that further relevant policy measures can be taken. First, based on the laws revealed by the quantitative verification of industrial transfer by environmental regulation, we should further guide the rational transfer of industrial groups in the Yangtze River economic belt with differentiated environmental regulations. Based on the Markov process model of industrial transfer established in this analysis, the process produces real-time research and judgment on the trend of industrial transfer between provinces and cities in the Yangtze River economic belt, and conducts Monte Carlo analysis on the industrial transfer situation caused by changes in the relative strength and weakness of inter-regional environmental regulation. Secondly, the quantitative results indicate that stable and sustainable environmental regulation should play a long-term role on environmental conditions. The numerical simulation and regression analysis in this analysis show that the cumulative amount of environmental regulation has the effect of driving industrial transfer and improving ecological efficiency, while the comparative analysis presented in this paper show that the final impact of immediate environmental regulation on ecological efficiency is not significant, so it can be inferred that its driving effect is also weak. In conclusion, to achieve the promotion of eco-efficiency by strengthening environmental regulation, emphasis should be placed on the long-term robustness and sustainability of environmental regulation. Finally, based on the significance analysis of the control variables discovered in the regression analysis, it is concluded that the structure of foreign investment as a catalyst should be optimized for the driving effect of environmental regulation. According to the numerical results of regression fitting described in previous sections, it can be seen that the two control variables of external dependence and the proportion of foreign direct investment have always been significant. It is indispensable in the characterization of the law of action of ecological efficiency. It can be seen from the regression expression that foreign investment has played a positive role in the direct effect of industrial transfer on ecological efficiency. Therefore, optimizing the structure of foreign investment will help to give full play to the driving effect of environmental regulation. However, the convenience of foreign direct investment in different regions in the upper, middle, and lower reaches of the Yangtze River economic belt is significantly different. Therefore, other inland areas should strengthen their ability to absorb the radiation effect of foreign trade resources along the coast and rivers, and should speed the construction of inland transportation, so as to better undertake the assistance and thrust of foreign investment in the low-carbon and green industrial transformation.

The case study of the Yangtze River economic belt can to some extent be meaningful in a global view, especially for developing countries and regions. Although for many developing countries there are not economic belts as large as Yangtze River in China, regarding the same issues, we may consider an entire country or certain regions containing several nations (some potential examples are listed in Table 13). Some policy suggestions should remain valid, or at least the same approaches of analysis between the triple aspects should be applicable and strategies designed accordingly, but political barriers may vary significantly. In some situations, political barriers should be considered as another important control variable added to our model in Section 4.

## Figures and Tables

**Figure 1 ijerph-19-10127-f001:**
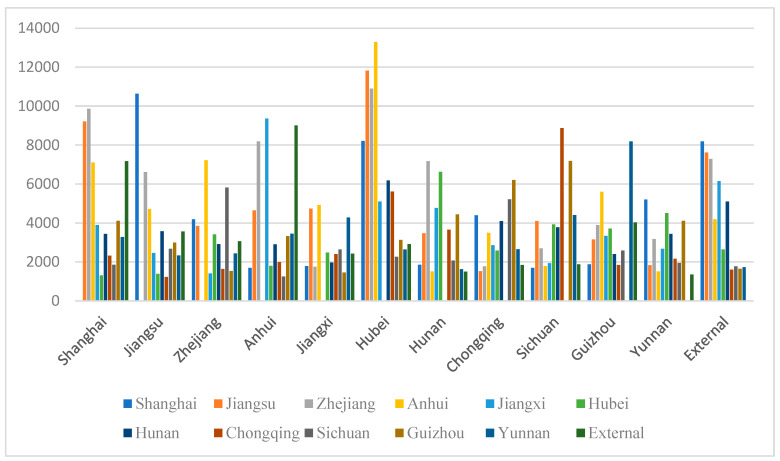
Cumulative directional transfer of industries in 11 provinces and cities in the Yangtze River economic belt.

**Figure 2 ijerph-19-10127-f002:**
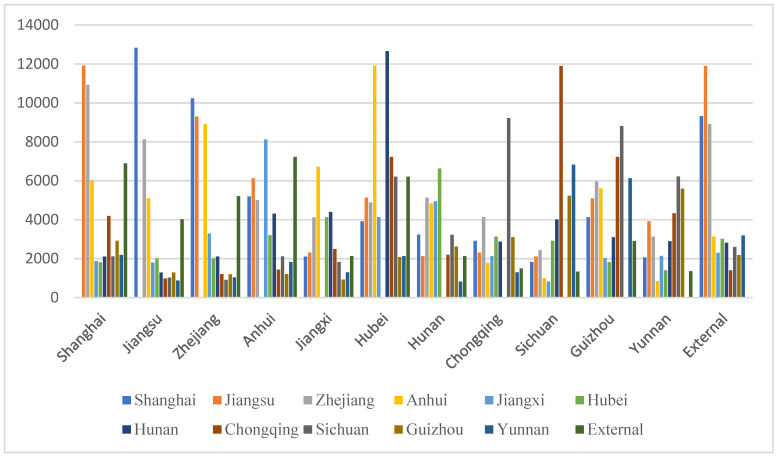
Cumulative directional transfer of industries in 11 provinces and cities in the Yangtze River economic belt.

**Figure 3 ijerph-19-10127-f003:**
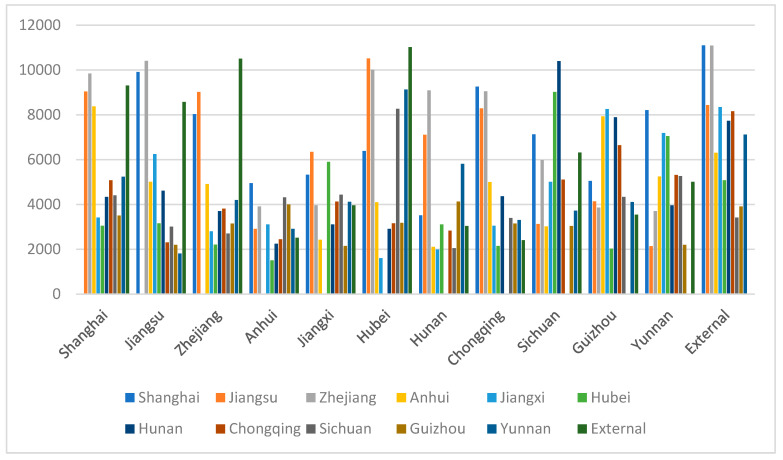
Cumulative directional transfer of industries in 11 provinces and cities in the Yangtze River economic belt.

**Table 1 ijerph-19-10127-t001:** Intensity of environmental regulation in provinces and cities of the Yangtze River economic belt: 2005–2018 (unit: percentage).

Region	2005	2006	2007	2008	2009	2010	2011	2012	2013	2014	2015	2016	2017	2018
Shanghai	0.21	0.13	0.11	0.18	0.13	0.14	0.09	0.16	0.07	0.24	0.30	0.69	0.53	0.09
Jiangsu	0.42	0.25	0.22	0.26	0.16	0.10	0.14	0.16	0.23	0.18	0.22	0.25	0.13	0.22
Zhejiang	0.31	0.33	0.28	0.14	0.18	0.09	0.12	0.18	0.35	0.40	0.34	0.32	0.19	0.17
Anhui	0.25	0.25	0.20	0.33	0.27	0.11	0.13	0.16	0.46	0.19	0.19	0.41	0.24	0.17
Jiangxi	0.50	0.38	0.30	0.18	0.12	0.15	0.12	0.07	0.24	0.18	0.21	0.14	0.14	0.25
Hubei	0.61	0.51	0.43	0.37	0.54	0.41	0.11	0.15	0.24	0.24	0.14	0.29	0.13	0.10
Hunan	0.64	0.64	0.51	0.34	0.28	0.22	0.12	0.20	0.23	0.16	0.24	0.11	0.07	0.06
Chongqing	0.38	0.30	0.23	0.48	0.24	0.21	0.11	0.08	0.15	0.10	0.11	0.06	0.09	0.08
Sichuan	0.79	0.65	0.52	0.39	0.17	0.10	0.18	0.10	0.16	0.20	0.11	0.10	0.11	0.13
Guizhou	0.83	1.18	1.00	0.82	0.71	0.45	0.72	0.56	0.73	0.59	0.32	0.15	0.13	0.16
Yunnan	0.57	0.67	0.55	0.50	0.45	0.41	0.46	0.57	0.63	0.63	0.56	0.33	0.15	0.22
Mean	0.50	0.48	0.40	0.36	0.30	0.22	0.21	0.22	0.32	0.28	0.25	0.26	0.17	0.15
SD	0.21	0.30	0.25	0.19	0.19	0.14	0.20	0.18	0.21	0.18	0.13	0.18	0.13	0.06
CV	0.41	0.62	0.63	0.53	0.65	0.64	0.95	0.81	0.65	0.63	0.53	0.69	0.74	0.43

Note: Data are calculated from the proportion of total investment in industrial pollution control in industrial added value. Data source: *China Statistical Yearbook* (2005–2018).

**Table 2 ijerph-19-10127-t002:** Increase and decrease in industrial transfer for 11 provinces and cities of the Yangtze River economic belt and other regions: 2005–2018.

	SH	JS	ZJ	AH	JX	HB	HN	CQ	SC	QZ	YN	External
2005	−294	292	−317	16	93	71	−41	−77	73	3	−124	305
2006	−203	−124	40	43	169	−16	68	26	128	−4	21	−148
2007	−314	−132	−11	112	75	83	134	−68	142	10	25	−56
2008	−523	−247	−469	149	48	140	243	−20	291	20	43	325
2009	−404	410	−380	331	138	569	234	165	445	25	−70	−1463
2010	62	−481	−144	462	452	502	440	−16	295	−70	116	−1618
2011	−532	−589	−412	577	349	574	548	220	147	68	−84	−866
2012	−598	−75	−515	364	10	567	312	309	305	224	246	−1149
2013	−374	129	56	312	267	−153	232	217	314	211	124	−1335
2014	−105	144	224	95	39	368	161	324	−188	164	−39	−1187
2015	−324	1269	612	−354	−1	347	461	177	−118	316	−97	−2288
2016	−120	476	232	255	76	393	−286	77	−322	193	−99	−875
2017	287	1063	−119	116	31	−50	−675	−167	−218	39	−67	−240
2018	196	110	265	260	−229	535	−628	−342	170	93	278	−708

Data source: *China Statistical Yearbook* (2005–2018).

**Table 3 ijerph-19-10127-t003:** Measurement results of eco-efficiency in the Yangtze River economic belt (input-oriented): 2005–2018.

Region	2005	2006	2007	2008	2009	2010	2011	2012	2013	2014	2015	2016	2017	2018
Shanghai	0.58	0.61	0.70	0.65	0.66	0.70	0.73	0.78	0.80	0.83	0.87	0.91	0.95	1.00
Jiangsu	0.88	0.48	0.46	0.47	0.47	0.48	0.49	0.50	0.51	0.52	0.55	0.57	0.59	0.62
Zhejiang	0.61	0.50	0.51	0.52	0.53	0.55	0.55	0.57	0.58	0.60	0.62	0.65	0.67	0.71
Anhui	0.77	0.37	0.35	0.32	0.31	0.30	0.30	0.30	0.31	0.32	0.33	0.34	0.35	0.37
Jiangxi	0.45	0.37	0.36	0.35	0.34	0.34	0.33	0.34	0.37	0.36	0.36	0.36	0.38	0.39
Hubei	0.37	0.36	0.35	0.35	0.34	0.33	0.33	0.33	0.34	0.35	0.36	0.38	0.39	0.41
Hunan	1.02	0.77	0.45	0.37	0.35	0.34	0.34	0.34	0.33	0.35	0.36	0.37	0.39	0.40
Chongqing	0.37	0.36	0.36	0.36	0.36	0.37	0.39	0.41	0.43	0.45	0.47	0.51	0.54	0.56
Sichuan	0.47	0.35	0.34	0.33	0.32	0.32	0.33	0.33	0.35	0.36	0.37	0.39	0.40	0.42
Guizhou	0.29	0.28	0.28	0.27	0.26	0.25	0.24	0.24	0.24	0.25	0.25	0.26	0.27	0.28
Yunnan	0.34	0.32	0.31	0.29	0.29	0.28	0.28	0.28	0.29	0.29	0.30	0.31	0.31	0.33

Data source: *China Statistical Yearbook* (2005–2018).

**Table 4 ijerph-19-10127-t004:** Cumulative directional transfer of industries in 11 provinces and cities in the Yangtze River economic belt: 2005–2018.

	SH	JS	ZJ	AH	JX	HB	HN	CQ	SC	QZ	YN
SH	0	2398	9862	2414	3888	1303	3442	2315	1853	4104	3273
JS	10,626	0	3427	1793	2454	1384	13,578	1229	2679	13,211	2333
ZJ	2631	3844	0	2014	1407	3413	2908	1641	5822	1533	2430
AH	1696	4643	2136	0	9367	1792	2899	1987	1250	3318	3443
JX	1784	4737	1754	3116	0	2485	1965	2397	2639	1460	4284
HB	2521	2490	2417	1388	1664	0	2095	5613	2265	3135	2645
HN	1851	3467	10,968	1517	4767	1218	0	3657	2074	4442	1624
CQ	4396	1527	1777	3492	2853	2582	4101	0	2371	3089	2656
SC	1690	4100	2691	1786	1936	3925	3782	2463	0	1426	4410
GZ	1871	3150	3894	15,061	3335	3706	2400	1839	2581	0	2282
YN	5207	1827	3176	1515	2677	4503	3433	2164	1942	2549	0

**Table 5 ijerph-19-10127-t005:** Cumulative directional transfer of industries in 11 provinces and cities in the Yangtze River economic belt (the distance influence intensity coefficient is set to 0.5).

	SH	JS	ZJ	AH	JX	HB	HN	CQ	SC	QZ	YN	EX
SH	0	9211	9862	7109	3888	1303	3442	2315	1853	4104	3273	7170
JS	10,626	0	6613	4718	2454	1384	3578	1229	2679	2991	2333	3562
ZJ	4192	3844	0	7221	1407	3413	2908	1641	5822	1533	2430	3059
AH	1696	4643	8192	0	9367	1792	2899	1987	1250	3318	3443	9004
JX	1784	4737	1754	4921	0	2485	1965	2397	2639	1460	4284	2422
HB	8212	11,812	10,891	13,291	5102	0	6188	5613	2265	3135	2645	2917
HN	1851	3467	7170	1517	4767	6626	0	3657	2074	4442	1624	1499
CQ	4396	1527	1777	3492	2853	2582	4101	0	5218	6201	2656	1845
SC	1690	4100	2691	1786	1936	3925	3782	8871	0	7181	4410	1878
GZ	1871	3150	3894	5601	3335	3706	2400	1839	2581	0	8191	4034
YN	5207	1827	3176	1515	2677	4503	3433	2164	1942	4110	0	1352
EX	8191	7622	7291	4191	6143	2643	5103	1608	1769	1653	1724	0

**Table 6 ijerph-19-10127-t006:** Cumulative directional transfer of industries in 11 provinces and cities in the Yangtze River economic belt (the distance influence intensity coefficient is set to 0.8).

	SH	JS	ZJ	AH	JX	HB	HN	CQ	SC	QZ	YN	EX
SH	0	11,912	10,921	6019	1882	1809	2102	4192	2122	2920	2188	6891
JS	12,829	0	8122	5102	1802	2012	1288	981	1022	1291	871	4022
ZJ	10,229	9290	0	8910	3291	2013	2102	1201	910	1199	1031	5210
AH	5192	6131	5002	0	8122	3201	4310	1429	2122	1202	1821	7219
JX	2109	2313	4121	6722	0	4121	4391	2491	1822	921	1292	2131
HB	3921	5121	4882	11,921	4121	0	12,661	7219	6212	2071	2128	6210
HN	3233	2129	5121	4829	4941	6626	0	2199	3219	2618	821	2125
CQ	2914	2313	4129	1772	2134	3129	2871	0	9218	3102	1302	1489
SC	1819	2102	2441	1004	823	2910	4014	11,892	0	5233	6821	1332
GZ	4121	5089	5952	5601	2031	1822	3092	7219	8802	0	6128	2901
YN	2055	3921	3129	839	2139	1396	2890	4329	6219	5592	0	1352
EX	9316	11,890	8912	3129	2303	3019	2811	1399	2600	2191	3186	0

**Table 7 ijerph-19-10127-t007:** Environmental regulation intensity matrix A after perturbation by the first type of transformation.

0.21	0.13	0.11	0.18	0.13	0.14	0.09	0.16	0.07	0.24	0.31	0.70	0.54	0.09
0.43	0.26	0.22	0.27	0.16	0.10	0.14	0.16	0.23	0.18	0.22	0.26	0.13	0.22
0.32	0.34	0.29	0.14	0.18	0.09	0.12	0.18	0.36	0.41	0.35	0.33	0.19	0.17
0.26	0.26	0.20	0.34	0.28	0.11	0.13	0.16	0.47	0.19	0.19	0.42	0.24	0.17
0.51	0.39	0.31	0.18	0.12	0.15	0.12	0.07	0.24	0.18	0.21	0.14	0.14	0.26
0.62	0.52	0.44	0.38	0.55	0.42	0.11	0.15	0.24	0.24	0.14	0.30	0.13	0.10
0.65	0.65	0.52	0.35	0.29	0.22	0.12	0.20	0.23	0.16	0.24	0.11	0.07	0.06
0.39	0.31	0.23	0.49	0.24	0.21	0.11	0.08	0.15	0.10	0.11	0.06	0.09	0.08
0.81	0.66	0.53	0.40	0.17	0.10	0.18	0.10	0.16	0.20	0.11	0.10	0.11	0.13
0.85	1.20	1.02	0.84	0.72	0.46	0.73	0.57	0.74	0.60	0.33	0.15	0.13	0.16
0.58	0.68	0.56	0.51	0.46	0.42	0.47	0.58	0.64	0.64	0.57	0.34	0.15	0.22
0.51	0.49	0.41	0.37	0.31	0.22	0.21	0.22	0.33	0.29	0.26	0.27	0.17	0.15

**Table 8 ijerph-19-10127-t008:** Environmental regulation intensity matrix A after perturbation by the second type of transformation.

0.42	0.25	0.22	0.26	0.16	0.10	0.14	0.16	0.23	0.18	0.22	0.25	0.13	0.22
0.21	0.13	0.11	0.18	0.13	0.14	0.09	0.16	0.07	0.24	0.30	0.69	0.53	0.09
0.25	0.25	0.20	0.33	0.27	0.11	0.13	0.16	0.46	0.19	0.19	0.41	0.24	0.17
0.31	0.33	0.28	0.14	0.18	0.09	0.12	0.18	0.35	0.40	0.34	0.32	0.19	0.17
0.61	0.51	0.43	0.37	0.54	0.41	0.11	0.15	0.24	0.24	0.14	0.29	0.13	0.10
0.50	0.38	0.30	0.18	0.12	0.15	0.12	0.07	0.24	0.18	0.21	0.14	0.14	0.25
0.38	0.30	0.23	0.48	0.24	0.21	0.11	0.08	0.15	0.10	0.11	0.06	0.09	0.08
0.64	0.64	0.51	0.34	0.28	0.22	0.12	0.20	0.23	0.16	0.24	0.11	0.07	0.06
0.83	1.18	1.00	0.82	0.71	0.45	0.72	0.56	0.73	0.59	0.32	0.15	0.13	0.16
0.79	0.65	0.52	0.39	0.17	0.10	0.18	0.10	0.16	0.20	0.11	0.10	0.11	0.13
0.50	0.48	0.40	0.36	0.30	0.22	0.21	0.22	0.32	0.28	0.25	0.26	0.17	0.15
0.57	0.67	0.55	0.50	0.45	0.41	0.46	0.57	0.63	0.63	0.56	0.33	0.15	0.22

**Table 9 ijerph-19-10127-t009:** Environmental regulation intensity matrix A after perturbation by the third type of transformation.

0.64	0.39	0.33	0.45	0.29	0.24	0.23	0.32	0.30	0.42	0.52	0.95	0.66	0.31
0.42	0.25	0.22	0.26	0.16	0.10	0.14	0.16	0.23	0.18	0.22	0.25	0.13	0.22
0.57	0.59	0.48	0.48	0.46	0.20	0.25	0.34	0.82	0.59	0.53	0.74	0.43	0.34
0.25	0.25	0.20	0.33	0.27	0.11	0.13	0.16	0.46	0.19	0.19	0.41	0.24	0.17
1.12	0.90	0.74	0.56	0.67	0.57	0.23	0.22	0.48	0.42	0.35	0.44	0.27	0.35
0.61	0.51	0.43	0.37	0.54	0.41	0.11	0.15	0.24	0.24	0.14	0.29	0.13	0.10
0.64	0.64	0.51	0.34	0.28	0.22	0.12	0.20	0.23	0.16	0.24	0.11	0.07	0.06
1.03	0.95	0.75	0.83	0.53	0.43	0.23	0.28	0.38	0.26	0.35	0.17	0.16	0.14
1.64	1.85	1.54	1.23	0.89	0.56	0.91	0.67	0.90	0.80	0.44	0.25	0.24	0.29
0.83	1.18	1.00	0.82	0.71	0.45	0.72	0.56	0.73	0.59	0.32	0.15	0.13	0.16
1.08	1.16	0.96	0.87	0.76	0.63	0.67	0.79	0.96	0.92	0.82	0.60	0.32	0.37
0.50	0.48	0.40	0.36	0.30	0.22	0.21	0.22	0.32	0.28	0.25	0.26	0.17	0.15

**Table 10 ijerph-19-10127-t010:** Cumulative directional transfer of industries in 11 provinces and cities in the Yangtze River economic belt.

	SH	JS	ZJ	AH	JX	HB	HN	CQ	SC	QZ	YN	EX
SH	0	8996	10,109	6509	2967	1489	2810	3057	2133	3524	2751	6909
JS	11,338	0	7114	4784	2131	1620	2544	1100	1929	3209	1672	3683
ZJ	5612	6054	0	7785	2170	2756	3019	1423	3642	1362	1793	3901
AH	3130	5175	6688	0	8656	2347	3433	1712	2011	2362	2692	7923
JX	1884	3624	2713	5579	0	3122	2944	2388	2243	2020	2941	2251
HB	6245	7912	8149	12,449	4587	0	8782	6175	3877	2625	3078	4129
HN	2394	2499	7212	2877	4744	6493	0	2971	2513	3587	1254	1731
CQ	3685	1826	2729	2699	2493	2760	3502	0	6793	4775	2034	3819
SC	2101	3178	3013	2044	2067	3420	3803	9962	0	6219	5334	1611
GZ	2778	4066	4680	5488	2720	2287	2642	4061	5142	0	7160	3477
YN	4019	2669	3092	1200	2397	3108	3136	3030	3699	4650	0	1009
EX	5101	6691	7826	3661	4407	2748	2873	1488	1031	2171	3012	0

**Table 11 ijerph-19-10127-t011:** Cumulative directional transfer of industries in 11 provinces and cities in the Yangtze River economic belt.

	SH	JS	ZJ	AH	JX	HB	HN	CQ	SC	QZ	YN	EX
SH	0	9751	10,073	6891	3486	1404	3174	2690	1906	3867	3056	7114
JS	11,066	0	6914	4794	2323	1509	3120	1179	2347	2651	2040	3654
ZJ	5399	4933	0	7558	1783	3133	2746	1553	4839	1466	2150	3489
AH	2395	4940	7554	0	9118	2073	3181	1875	1424	2894	3118	8647
JX	1849	4252	2227	5281	0	2812	2450	2415	2475	1352	3685	2363
HB	7353	10,473	9689	13,017	4905	0	7482	5934	3054	2922	2541	3575
HN	2127	3199	6760	2179	4801	6626	0	3365	2303	4077	1463	1624
CQ	4099	1684	2247	3148	2709	2691	3855	0	6018	5581	2385	1773
SC	1715	3700	2641	1629	1713	3722	3828	9475	0	6791	4892	1768
GZ	2321	3537	4305	5601	3074	3329	2538	2915	3825	0	7778	3807
YN	4576	2245	3166	1379	2569	3881	3324	2597	2797	4406	0	1352
EX	8416	8475	7615	3978	5375	2718	4644	1566	1935	1760	2016	0

**Table 12 ijerph-19-10127-t012:** Cumulative directional transfer of industries in 11 provinces and cities in the Yangtze River economic belt.

	SH	JS	ZJ	AH	JX	HB	HN	CQ	SC	QZ	YN	EX
SH	0	8941	9699	8317	3200	1190	4061	3192	2089	3994	3811	6982
JS	9868	0	9772	6198	3290	2319	3033	1329	2515	3012	3144	2791
ZJ	5129	4021	0	9712	2432	3866	2791	1822	4795	1990	2719	4080
AH	2103	5311	8517	0	8869	1913	3160	2233	2069	2991	4012	8277
JX	1909	5229	2319	5672	0	3001	1734	2619	2388	1913	3709	3081
HB	7812	12,017	10,033	12,660	3921	0	4612	6712	6102	3129	4924	2371
HN	2134	3913	4901	3216	3912	4551	0	5182	3328	3235	2048	2190
CQ	5013	1203	2301	2846	3011	3398	3921	0	4961	7392	3014	2051
SC	2181	3917	3081	2781	3012	4013	3191	10,212	0	9123	4910	2301
GZ	3001	2889	3891	4981	4239	4166	2188	2161	3078	0	7731	3977
YN	4129	2008	2969	1867	3081	4227	4062	3757	2810	4671	0	2775
EX	6879	8170	7305	4710	5003	3035	4912	1558	1914	1992	2048	0

**Table 13 ijerph-19-10127-t013:** Zones of economic cooperation identified as potential pilots for applying our analysis.

Zone of Economic Cooperation	Participants
Greater Mekong Subregional Economic Cooperation Zone	China, Myanmar, Laos, Thailand, Cambodia and Vietnam
South Asian Association for Regional Cooperation Zone	Bangladesh, Bhutan, India, Maldives, Nepal, Pakistan, Sri Lanka, Afghanistan
Union of the Arab Maghreb	Libya, Tunisia, Algeria, Morocco, Mauritania
Economic Community of West African States	Libya, Tunisia, Algeria, Morocco, Mauritania are Benin, Mali, Niger, Mauritania, Senegal, Cote d‘Ivoire, Burkina Faso
Southern Cone Common Market	Argentina, Brazil, Paraguay, Uruguay

## Data Availability

https://kns.cnki.net/kns8/defaultresult/index (accessed on 2 July 2022).

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
