# Peer review of "Environmental Regulation Promotes Eco-Efficiency through Industrial Transfer: Evidence from the Yangtze River Economic Belt in China"

_ijerph, 2022, doi:10.3390/ijerph191610127_

Round 1

Reviewer 1 Report

The authors incorporate a topical study that seeks to understand the implication that specific environmental policies can have on eco-efficiency. Specifically, they focus on establishing the real power of industrial transfer in the system described above.

The authors focus on a specific region of China to develop their work. The development of the work is done taking into account Markov processes, specifically to establish the relationships concerning the industrial transfer.

In addition to the aforementioned study and before moving on to the major and minor revision needs, it is essential to highlight that the study lacks a global sense; that is, global and strategic key concepts are worked on and developed based on a Chinese region and the data from this region (it is challenging to establish conclusions for a global world based on a specific area of the world, even if it is precious, such as the Yangtze River region).

Major revisions:
The main criticism has already been stated; that is, it does not appear to be a generalisable study. The title of the study is undoubtedly ambitious, and it is not clear that it has a clear reflection on the real science described.

It is suggested to the authors to change the whole background of the study by focusing it on a specific region, on a particular case. Otherwise, this article cannot be accepted for publication.

In the introduction, the authors allude to the Environmental Kuznets curve, and it is therefore considered necessary to incorporate literature on this subject.

There are different gaps in the literature, such as in point 2.2 and carbon emissions.

Point 2.3 incorporates different concepts, such as product life cycle theory or marginal industry expansion theory, on which more needs to be done. It is not possible to include so many concepts without being careful about their justification.

The first paragraph of point 2.4 needs further elaboration. It is a very profound statement.

More justification is needed for the variables introduced in point 3.1.
Although usually those related to the literature are highlighted as minor reviews, the reviewer would like to point out that not only has he found indeed old studies such as those indicated in 1999 and 2000 but also that these studies are not located in the literature. Aitken is cited in 1999 but is not found in the bibliography (not an isolated case).

Minor revisions:
The authors mention table 2.2, which is not observed in the text.

A citation error is detected in the text, such as "... Zhang jun, gui-ying wu (2004)...".
The formatting should be taken care of and not leave so much blank space on page 15.
Correctly indicate the image "Code 6-3".

Author Response

See the word file attached.

Reviewer 2 Report

My comments are the follows:

- the paper always overlaps methods and results. The case study is presented by the first lines of introduction and it is not easy to understand what the specific of the case stusy application, and what the general context. 

chapter 1 provides a kind of literature review for the three main topics of the research. The review is confused and unefficient for whom (as my case) are not expert of the specific topic of the research. I suggest to state the refernces adopted, and what the criteria for the choice.

chapter 3 is a key part of the paper, but actually it is not clear.

There is a general lack about the data source. 

I suggest to separate the methods form the results. Statements about the research assumptions is reccomended. Conclusion must be clearly addressed to the research questions.

Author Response

Please receive the word-file attached.

Round 2

Reviewer 1 Report

A document with the revisions is attached

Reviewer 2 Report

Authors improve the previous text in terms of scientific standards. I suggest to verify the refences in section 2
